# Learning Better Representations From Less Data For Propositional Satisfiability

**Mohamed Ghanem**[*]    **Frederik Schmitt**[*]    **Julian Siber**[*]    **Bernd Finkbeiner**[*]

[*]CISPA Helmholtz Center for Information Security

{mohamed.ghanem,frederik.schmitt,julian.siber,finkbeiner}@cispa.de

## Abstract

Training neural networks on NP-complete problems typically demands very large amounts of training data and often needs to be coupled with computationally expensive symbolic verifiers to ensure output correctness. In this paper, we present NeuRes, a neuro-symbolic approach to address both challenges for propositional satisfiability, being the quintessential NP-complete problem. By combining certificate-driven training and expert iteration, our model learns better representations than models trained for classification only, with a much higher data efficiency – requiring orders of magnitude less training data. NeuRes employs propositional resolution as a proof system to generate proofs of unsatisfiability and to accelerate the process of finding satisfying truth assignments, exploring both possibilities in parallel. To realize this, we propose an attention-based architecture that autoregressively selects pairs of clauses from a dynamic formula embedding to derive new clauses. Furthermore, we employ expert iteration whereby model-generated proofs progressively replace longer teacher proofs as the new ground truth. This enables our model to reduce a dataset of proofs generated by an advanced solver by $\sim 32\%$ after training on it with no extra guidance. This shows that NeuRes is not limited by the optimality of the teacher algorithm owing to its self-improving workflow. We show that our model achieves far better performance than NeuroSAT in terms of both correctly classified and proven instances.

## 1    Introduction

Boolean satisfiability (SAT) is a fundamental problem in computer science. For theory, this stems from SAT being the first problem proven NP-complete [13]. For practice, this is due to many highly-optimized SAT solvers being used as flexible reasoning engines in a variety of tasks such as model checking [12, 47], software verification [17], planning [27], and mathematical proof search [24]. Recently, SAT has also served as a litmus test for assessing the symbolic reasoning capabilities of neural models and a promising domain for neuro-symbolic systems [43, 42, 1, 10, 36]. So far, neural models only provide limited, if any, justification for unsatisfiability predictions. NeuroCore [42], for example, predicts an unsatisfiable core, the verification of which can be as hard as solving the original problem. No certificates at all or certificates that are hard to check limit neural methods in a domain where correctness is critical and prevents close integrations with symbolic methods. Therefore, we propose a neuro-symbolic model that utilizes *resolution* to solve SAT problems by generating easy-to-check certificates.

A resolution proof is a sequence of case distinctions, each involving two clauses, that ends in the empty clause (falsum). This technique can also be used to prove satisfiability by exhaustively applying it until no further new resolution steps are possible and the empty clause has not been derived. Generating such a proof is an interesting problem from a neuro-symbolic perspective because unlike other discrete combinatorial problems that have been considered before [46, 7, 28, 30, 11], it

requires selecting compatible *pairs* of clauses from the dynamically growing pool, as newly derived clauses are naturally considered for derivation in subsequent steps. In this work, we devise three attention-based mechanisms to perform this pair-selection needed for generating resolution proofs. In addition, we augment the architecture to efficiently handle *sat* (satisfiable) formulas with an assignment decoding mechanism that assigns a truth value to each literal. We hypothesize that, despite their final goals being in complete opposition, resolution and *sat* assignment finding can form a mutually beneficial collaboration. On the one hand, clauses derived by resolution incrementally inject additional information into the network, e.g., deriving a single-literal clause by resolution directly implies that literal should be true in any possible *sat* assignment. On the other hand, finding a *sat* assignment absolves the resolution network from having to prove satisfiability by exhaustion. On that basis, given an input formula, NeuRes proceeds in two parallel tracks: (1) finding a *sat* assignment, and (2) deriving a resolution proof of unsatisfiability. Both tracks operate on a shared representation of the problem state. Depending on which track succeeds, NeuRes produces the corresponding SAT verdict which is guaranteed to be sound by virtue of its certificate-based design. Since both of our certificate types are efficient to check, we can afford to perform these symbolic checks at each step. When comparing NeuRes with NeuroSAT [43], which has been trained to predict satisfiability with millions of samples, we demonstrate that NeuRes achieves a higher accuracy while providing a proof and requires only thousands of training samples.

As for most problems in theorem proving we are not only interest in finding any proof but a short proof. Resolution proofs can vary largely in their size depending on the resolution steps taken. Being able to efficiently check the proof, also allows us to adapt the proof target while training the model. In particular, we explore an expert iteration mechanism [2, 39] that pre-rolls the resolution proof of the model and replaces the target proof whenever the pre-rolled proof is shorter. We demonstrate that this bootstrapping mechanism iteratively shortens the proofs of our training dataset while further improving the overall performance of the model.

We make the following contributions:

1. We introduce novel architectures which combine graph neural networks with attention mechanisms for generating resolution proofs and assignments for CNF formulas (Section 4).

2. We show that for propositional logic, learning to prove rather than predict satisfiability results in better representations and requires far less training samples (Section 6 and 7).

3. We devise a bootstrapped training procedure where our model progressively produces shorter resolution proofs than its teacher (Section 6.2) boosting the model's overall performance.

The implementation of our framework can be found at `https://github.com/Oschart/NeuRes`.

## 2  Related Work

**SAT Solving and Certificates.**  We refer to the annual SAT competitions [5] for a comprehensive overview on the ever-evolving landscape of SAT solvers, benchmarks, and proof checkers. SAT solvers are complex systems with a documented history of bugs [9, 26], hence proof certificates have been partially required in this competition since 2013 [3]. Unlike satisfiable formulas, there are several ways to certify unsatisfiable formulas [23]. Resolution proofs [52, 20] are easy to verify [15], but non-trivial to generate from modern solvers based on the paradigm of conflict-driven clause learning [34]. Clausal proofs, e.g., in DRAT format [50], are easier to generate and space-efficient, but hard to validate. Verifying the proofs can take longer than their discovery [22] and requires highly optimized algorithms [31].

**Deep Learning for SAT Solving.**  NeuroSAT [43] was the first study of the Boolean satisfiability problem as an end-to-end learning task. Building upon the NeuroSAT architecture, a simplified version has been trained to predict unsatisfiable cores and successfully integrated as a branching heuristic in a state-of-the-art SAT solver [42]. Recent work has employed a related architecture as a phase selection heuristic [49]. It has been shown that both the NeuroSAT architecture and a newly introduced deep exchangeable architecture can outperform SAT solvers on instances of 3-SAT problems [10]. The NeuroSAT architecture has also been applied on special classes of crypto-analysis problems [44]. In addition to supervised learning, unsupervised methods have been proposed for solving SAT problems. For Circuit-SAT a deep-gated DAG recursive neural architecture has been

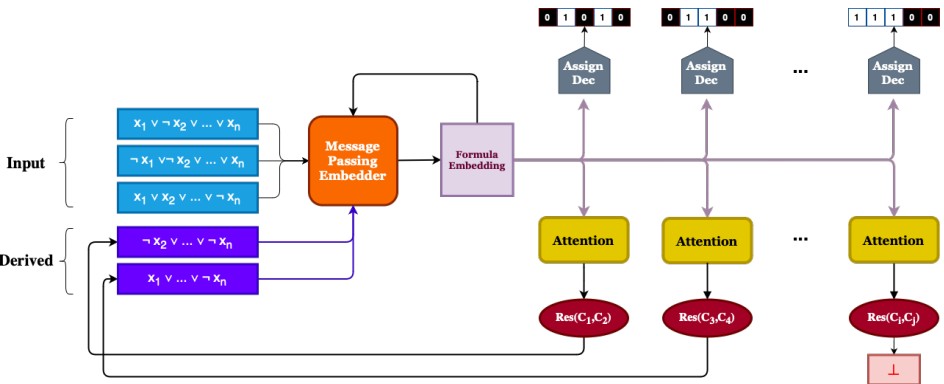

Figure 1: Overall NeuRes architecture

presented together with a differentiable training objective to optimize towards solving the Circuit-SAT problem and finding a satisfying assignment [1]. For Boolean satisfiability, a differentiable training objective has been proposed together with a query mechanism that allows for recurrent solution trials [36].

**Deep Learning for Formal Proof Generation.** In formal mathematics, deep learning has been integrated with theorem proving for clause selection [33, 18], premise selection [25, 48, 6, 35], tactic prediction [51, 37] and whole proof searches [40, 19]. For SMT formulas specifically, deep reinforcement learning has been applied to tactic prediction [4]. In the domain of quantified boolean formulas, heuristics have been learned to guide search algorithms in proving the satisfiability and unsatisfiability of formulas [32]. For temporal logics, deep learning has been applied to prove the satisfiability of linear-time temporal logic formulas and the realizability of specifications [21, 41, 14].

## 3 Proofs of (Un-)Satisfiability

We start with a brief review of certifying the (un-)satisfiability of propositional formulas in conjunctive normal form. For a set of Boolean variables $V$, we identify with each variable $x \in V$ the *positive literal* $x$ and the *negative literal* $\neg x$ denoted by $\bar{x}$. A *clause* corresponds to a disjunction of literals and is abbreviated by a set of literals, e.g., $\{\bar{1}, 3\}$ represents $(\neg x_1 \vee x_3)$. A formula in *conjunctive normal form (CNF)* is a conjunction of clauses and is abbreviated by a set of clauses, e.g., $\{\{\bar{1}, 3\}, \{1, 2, \bar{4}\}\}$ represents $(\neg x_1 \vee x_3) \wedge (x_1 \vee x_2 \vee \neg x_4)$. Any Boolean formula can be converted to an equisatisfiable CNF formula in polynomial time, for example with Tseitin transformation [45].

A CNF formula is *satisfiable* if there exists an *assignment* $\mathcal{A} : V \rightarrow \{\top, \bot\}$ such that all clauses are satisfied, i.e., each clause contains a positive literal $x$ such that $\mathcal{A}(x) = \top$ or a negative literal $\bar{x}$ such that $\mathcal{A}(x) = \bot$. If no such assignment exists we call the formula *unsatisfiable*. To prove unsatisfiability we rely on resolution, a fundamental inference rule in satisfiability testing [16]. The resolution rule (*Res*) picks clauses with two opposite literals and performs the following inference:

$$\frac{C_1 \cup \{x\} \quad C_2 \cup \{\bar{x}\}}{C_1 \cup C_2} \; Res$$

Resolution effectively performs a case distinction on the value of variable $x$: Either it is assigned to $false$, then $C_1$ has to evaluate to $true$, or it is assigned to $true$, then $C_2$ has to evaluate to $true$. Hence, we may infer the clause $C_1 \cup C_2$. A *resolution proof* for a CNF formula is a sequence of applications of the *Res* rule ending in the empty clause.

## 4 Models

### 4.1 General Architecture

NeuRes is a neural network that takes a CNF formula as a set of clauses and outputs either a satisfying truth assignment or a resolution proof of unsatisfiability. As such, our model comprises a formula

embedder connected to two downstream heads: (1) an attention network responsible for selecting clause pairs, and (2) a truth assignment decoder. See Figure 1 for an overview of the NeuRes architecture. After obtaining the initial clause and literal embeddings (representing the input formula), we continue with the iterative certificate generation phase. At each step, the model selects a clause pair which gets resolved into a new clause to append to the current formula graph while decoding a candidate truth assignment in parallel. The model keeps deriving new clauses until the empty clause is found (marking resolution proof completion), a satisfying assignment is found (marking a certified *sat* verdict), or the limit on episode length is reached (marking timeout).

## 4.2 Message-Passing Embedder

Similar to NeuroSAT, we use a message-passing GNN to obtain clause and literal embeddings by performing a predetermined number of rounds. Our formula graph is also constructed in a similar fashion to NeuroSAT graphs where clause nodes are connected to their constituent literal nodes and literals are connect to their complements (cf. Appendix A). For a formula in $m$ variables and $n$ clauses, the outputs of this GNN are two matrices: $E^L \in \mathbb{R}^{m \times d}$ for literal embeddings and $E^C \in \mathbb{R}^{n \times d}$ for clause embedding, where $d \in \mathbb{N}^+$ is the embedding dimension. Here we have two key differences from NeuroSAT. Firstly, NeuroSAT uses these embeddings as voters to predict satisfiability through a classification MLP. In our case, we use these embeddings as clause tokens for clause pair selection and literal tokens for truth value assignment. Secondly, since our model derives new clauses with every resolution step, we need to embed these new clauses, as well as update existing embeddings to reflect their relation to the newly inferred clauses. Consequently, we need to introduce a new phase to the message-passing protocol, for which we explore two approaches: *static embeddings* and *dynamic embeddings*.

In a *static* approach, we do not change the embeddings of initial clauses upon inferring a new clause. Instead, we exchange local messages between the node corresponding to the new clause and its literal nodes, in both directions. The main advantage of this approach is its low cost. A major drawback is that initial clauses never learn information about their relation to newly inferred clauses.

In a *dynamic* approach, we do not only generate a new clause and its embedding, we also update the embeddings of all other clauses. This accounts for the fact that the utility of an existing clause may change with the introduction of a new clause. We perform one message-passing round on the mature graph for every newly derived clause, which produces the new clause embedding and updates other clause embeddings. Since message-passing rounds are parallel across clauses, a single update to the whole embedding matrix is reasonably efficient.

## 4.3 Selector Networks

After producing clause and literal embeddings, NeuRes enters the derivation stage. At each step, our model needs to select two clauses to resolve, produce the resultant clause, and add it to the current formula. To realize our clause-pair selection mechanism, we employ three attention-based designs.

### 4.3.1 Cascaded Attention (Casc-Attn)

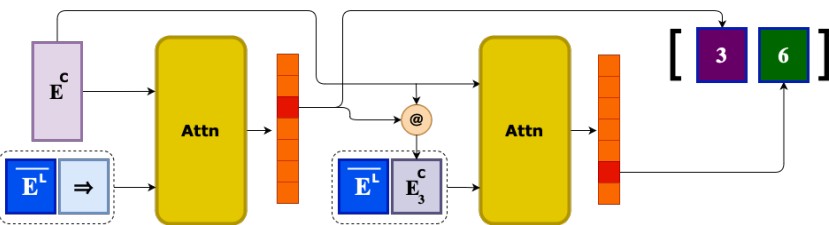

Figure 2: Cascaded attention

In this design, pairs are selected by making two consecutive attention queries on the clause pool. We condition the second attention query on the outcome (i.e., the clause) of the first query. Figure 2 shows this scheme where we perform the first query using the mean of the literal embeddings $\overline{E^L}$ concatenated with a zero vector while performing the second query using the mean of the literal

embeddings concatenated with the embedding vector $E_{c_1}^C$ of the clause selected in the first query. Formally, Casc-Attn selects a clause index pair $(c_1, c_2)$ as follows:

$$c_i = \underset{j}{\mathbf{argmax}} \left[ u^T \tanh(W_1 q_i + W_2 E_j^C) \right] \quad \text{with} \quad q_i = \begin{cases} \overline{E^L} \parallel \mathbf{0} & \text{if } i = 1 \\ \overline{E^L} \parallel E_{c_1}^C & \text{if } i = 2 \end{cases} \tag{1}$$

where $W_1 \in \mathbb{R}^{2d \times d}, W_2 \in \mathbb{R}^{d \times d}, u \in \mathbb{R}^d$ are trainable network parameters.

The advantage of this design is that it is not limited to pair selection and can be used to select a tuple of arbitrary length. The main downside, however, is that this design chooses $c_1$ independently from $c_2$, which is undesirable because the utility of a resolution step is determined by both clauses simultaneously (not sequentially).

### 4.3.2 Full Self-Attention (Full-Attn)

To address the downside of independent clause selection, this variant performs self-attention between all clauses to obtain a matrix $S \in \mathbb{R}^{n \times n}$ where $S_{i,j}$ represents the attention score of the clause pair $(c_i, c_j)$ as shown in Figure 3. The model selects clause pairs by choosing the cell with the maximal score. In this attention scheme, the clause embeddings are used as both queries and keys.

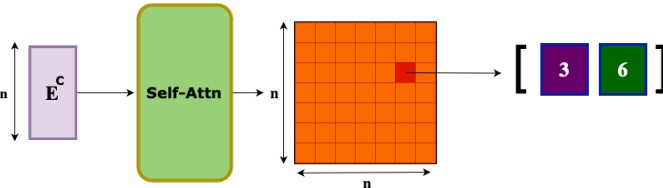

Figure 3: Full self-attention

Formally, Full-Attn selects a clause index pair $(c_1, c_2)$ as follows:

$$(c_1, c_2) = \underset{(i,j)}{\mathbf{argmax}} \ S_{i,j} \quad \text{with} \quad Q = E^C W_Q; \quad K = E^C W_K; \quad S = \frac{QK^T}{\sqrt{d}} \tag{2}$$

where $W_Q \in \mathbb{R}^{d \times d}, W_K \in \mathbb{R}^{d \times d}$ are trainable network parameters. Since $S$ contains many cells that correspond to invalid resolution steps (i.e., clause pairs that cannot be resolved), we mask out the invalid cells from the attention grid in ensure the network selection is valid at every step.

### 4.3.3 Anchored Self-Attention (Anch-Attn)

In Full-Attn, the attention grid grows quadratically with the number of clauses. In this variant, we relax this cost by exploiting a property of binary resolution where each step targets a single variable in the two resolvent clauses. This allows us to narrow down candidate clause pairs by first selecting a variable as an anchor on which our clauses should be resolved. As such, we do not need to consider the full clause set at once, only the clauses containing the chosen variable $v$. We further compress the attention grid by lining clauses containing the literal $v$ on rows while lining clauses containing the literal $\neg v$ on columns. This reduces the redundancy of the attention grid since clauses containing the variable $v$ with the same parity cannot be resolved on $v$, so there is no point in matching them. In this scheme, we have two attention modules: one attention network to choose an anchor variable followed by a self-attention network to produce the anchored score grid.

In light of Figure 4, this approach combines structural elements from Casc-Attn and Full-Attn; however, both elements are used differently in Anch-Attn. Firstly, the attention mechanism in Casc-Attn is used to select clauses whereas Anch-Attn uses it to select variables. Secondly, self-attention in Full-Attn matches any pair of clauses $(c_i, c_j)$ in both directions as the row and column dimensions in the attention score grid reflect the same clauses (all clauses). By contrast, Anch-Attn computes self-attention scores for clause pairs in only one order (positive instance to negative instance). Formally,

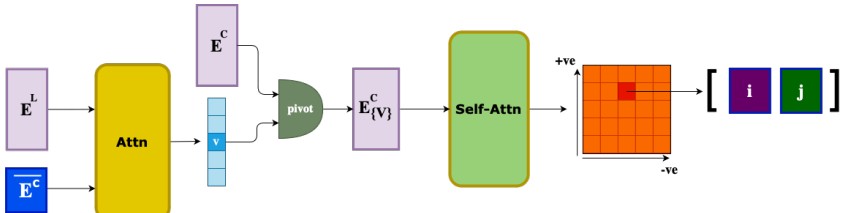

Figure 4: Anchored self-attention

Anch-Attn selects an anchor variable $v$ as follows:

$$v = \underset{i}{\textbf{argmax}} \left[ u^T \tanh(W_1 \overline{E^C} + W_2(E_i^{L^+} + E_i^{L^-})) \right] \tag{3}$$

where $W_1 \in \mathbb{R}^{d \times d}, W_2 \in \mathbb{R}^{d \times d}, u \in \mathbb{R}^d$ are trainable network parameters. The clause index pair $(c_1, c_2)$ is then selected according to the same equations of Full-Attn (Eq. 2) using the $v$-anchored set of clause embeddings.

### 4.4 Assignment Decoder

To extract satisfying assignments, we use a sigmoid-activated MLP $\psi$ on top of the literal embeddings $E^L$ to assign a truth value $\hat{\mathcal{A}}(l_i)$ to a literal $l_i$ as shown in Eq. 4.

$$\hat{\mathcal{A}}(l_i) = \sigma(\psi(E_i^L)) \tag{4}$$

Note that since for each variable, we have a positive and a negative literal embeddings, we can construct two different truth assignments at a time using this method. However, supervising both assignments did not improve the performance compared to only supervising the positive assignment (on positive literal embeddings). Thus, to simplify our loss function, we only derive truth assignments from the positive literal embeddings at train time while extracting both at test time. Interestingly, at test time, we found using negative literals (in addition to the positive ones) sometimes produces satisfying assignments before the positive branch despite receiving no direct supervision during training. Our intuition regarding this observation attributes it to the fact that the formula graph has no explicit notion of positive and negative literals, it only represents connections to clauses (positive and negative literals are connected by an undirected edge that does not distinguish their parity). As such, both literal nodes have a different local view into the rest of formula, which could result in one of them leading to a satisfying assignment faster than the other.

## 5 Training and Hyperparameters

### 5.1 Dataset

For our training and testing data, we adopt the same formula generation method as NeuroSAT [43], namely $\textbf{SR}(n)$ where $n$ is the number of variables in the formula. This method was designed to generate a generalized formula distribution that is not limited to a particular domain of SAT problems. To control our data distributions, we vary the range on the number of Boolean variables involved in each formula. For our training data, we use formulas in $\textbf{SR}(U(10, 40))$ where $U(10, 40)$ denotes the uniform distribution on integers between 10 and 40 (inclusive). To generate our teacher certificates comprising resolution proofs and truth assignments, we use the BooleForce solver [8] on the formulas generated on the $\textbf{SR}$ distribution.

### 5.2 Loss Function

We train our model in a supervised fashion using teacher-forcing on solver certificates. During *unsat* episodes, teacher actions (clause pairs) are imposed over the whole run. The length of the teacher proof dictates the length of the respective episode, denoted as $T$. Model parameters $\theta$ maximize the likelihood of teacher choices $y_t$ thereby minimizing the resolution loss $\mathcal{L}_{Res}$ shown in Eq 5.

$$\mathcal{L}_{Res} = -\frac{1}{T} \sum_t \log(p(y_t; \theta)) \cdot \gamma^{(T-t)} \tag{5}$$

During *sat* episodes, we minimize $\mathcal{L}_{sat}$ computed as the binary cross-entropy loss between the sigmoid-activated outputs of assignment decoder $\hat{\mathcal{A}} : V \to [0,1]$ and the teacher assignment $\mathcal{A} : V \to \{0,1\}$ as shown in Eq. 6.

$$\mathcal{L}_{sat} = \frac{1}{T} \sum_t \left[ \frac{\gamma^{(T-t)}}{|V|} \sum_v^V \text{BCE}(\hat{\mathcal{A}}(v), \mathcal{A}(v)) \right] \tag{6}$$

In both types of episodes, step-wise losses are weighted by a time-horizon discounting factor $\gamma < 1.0$ over the whole episode. The main rationale behind this is that later losses should have higher weights as the formula tends to get easier to solve with each new clause inferred by resolution.

### 5.3 Hyperparameters

NeuRes has several hyperparameters that influence network size, depth, and loss weighting. In the experiments we fix the embedding dimension to 128. We train our models with a batch size of 1 and the Adam optimizer [29] for 50 epochs which took about six days on a single NVIDIA A100 GPU. We linearly anneal the learning rate from $5 \times 10^{-5}$ to zero over the training episodes. This empirically yields better results than using a constant learning rate. We use a time discounting factor $\gamma = 0.99$ for the episodic loss. We apply global-norm gradient clipping with a ratio of 0.5 [38].

## 6 Generating Resolution Proofs

NeuRes uses resolution as the core reasoning technique for certificate generation, both in the *unsat* and *sat* cases. Hence, we start with an in-depth comparative evaluation of several internal variants for resolution only. In particular, we evaluate the success rate (i.e., problems solved before timeout) and proof length relative to the teacher, denoted by p-Len $= \frac{|\mathcal{P}_{\text{NeuRes}}|}{|\mathcal{P}_{\text{teacher}}|}$. We use a limit of 4 on this ratio as a timeout to avoid simply brute-forcing a resolution proof. Note that we measure p-Len only for solved formulas to avoid diluting the average with resolution trails that timed out. For experiments in this section, we train on 8K *unsat* formulas in $\mathbf{SR}(U(10, 40))$ and test our models on 10K unseen formulas belonging to the same distribution. We use more formulas than the model was trained on to more reliably demonstrate its learning capacity.

Table 1: Performance of all attention variants on *unsat* $\mathbf{SR}(U(10, 40))$ test problems.

| VARIANT | STATIC-EMBED | | DYNAMIC-EMBED | |
|---|---|---|---|---|
| | PROVEN (%) | P-LEN | PROVEN (%) | P-LEN |
| CASC-ATTN | 14.72 | 1.87 | 37.33 | 1.79 |
| FULL-ATTN | 25.38 | **1.61** | **95.2** | **1.67** |
| ANCH-ATTN | **28.72** | 2.12 | 60.5 | 2.28 |

### 6.1 Attention Variants

To assess the basic resolution performance of NeuRes, we evaluate each attention variant using both static and dynamic embeddings. For this experiment, we perform 32 rounds of message-passing for each input formula. As shown in Table 1, dynamic embedding is decisively better for all three attention variants, thereby confirming its conceptual merit. While anchored attention leads over other variants under static embeddings, full attention performs significantly better for dynamic embeddings, albeit at the cost of longer proofs on average. We believe that Anch-Attn's better performance in the static setting can be explained through the full connectivity of its attention grid (proven in Appendix B). Since dynamic-embedding Full-Attn is the best-performing configuration over in-distribution test settings, we will demonstrate the remaining evaluation experiments exclusively on this variant.

Table 2: Bootstrapped training data reduction statistics. Reduction statistics are computed on the $\mathbf{SR}(U(10, 40))$ training set while p-Len and success rate are computed on a test set of the same distribution.

| | |
|---|---|
| REDUCTION DEPTH | MAX: 23, AVG: 6.6 |
| PROOF REDUCTION (%) | MAX: 86.11, AVG: 33.51 |
| PROOFS REDUCED (%) | 90.08 |
| TOTAL REDUCTION (%) | 31.85 |
| P-LEN | 1.15 |
| SUCCESS RATE (%) | 100.0 |

## 6.2 Shortening Teacher Proofs with Bootstrapping

During our initial experiments, we discovered proofs produced by NeuRes that were shorter than the corresponding teacher proofs in the training data. Although teacher proofs were generated by a traditional SAT solver, they are not guaranteed to be size-optimal. The size of resolution proofs is their only real drawback, hence any method that can reduce this size would be immensely useful. Upon closer inspection we find that, on average, our previous best performer trained with regular teacher-forcing manages to shorten $\sim 18\%$ of teacher proofs by a notable factor (cf. Appendix C). This inspired us to devise a bootstrapped training procedure to capitalize on this feature: We pre-roll each input problem using model actions only, and whenever the model proof is shorter than the teacher's, it replaces the teacher's proof in the dataset. In other words, we maximize the likelihood of the shorter proof. In doing so iteratively, the model progressively becomes its own teacher by exploiting redundancies in the teacher algorithm.

The outcome of this bootstrapped training process is summarized in Table 6. We find that bootstrapping results in notable gains in terms of both success rate and optimality. The sharp decline in proof length (relatively quantified by p-Len) at test time shows that the models transfers the bootstrapped knowledge to unseen test formulas, as opposed to merely overfitting on training formulas. In addition to success rate and p-Len, we inspect the reduction statistics of our bootstrapped variant (first three rows of Table 6). Since the bootstrapped model performs multiple reduction scans over the training dataset, we add a metric for reduction depth computed as the number of progressive reductions made to a proof. To further quantify this effect, we report the maximum and average reduction ratios of reduced proofs relative to teacher proofs. Finally, we report the total reduction made to the dataset size in terms of total number of proof steps.

In Appendix C, we have compiled additional statistics (cf. Table 5) on proof shortening during the training process, as well as an example proof reduced by the bootstrapped NeuRes (Figure 7). We only include a small reduction example (from 20 steps to 10 steps) for space constraints, but we observed many more examples of much larger reductions (e.g., over $400$ steps).

## 7 Resolution-Aided SAT Solving

In this section, we evaluate the performance of our fully integrated model trained on a hybrid dataset comprising 8K unsatisfiable formulas (and their resolution proofs) and 8K satisfiable formulas (and their satisfying assignments). For the unsatisfiable formulas, timeout ($4 \times |\mathcal{P}_{\text{teacher}}|$) and optimality (p-Len) are measured similarly to previous experiments. For satisfiable formulas, we set the timeout (maximum #trials) to $2 \times |V|$. Ultimately, this section aims to investigate the effect of incorporating a certificate-driven downstream head on the quality of the learnt representations through its impact on the performance of the complementary task, i.e., proving/predicting satisfiability. We use NeuroSAT as our baseline as it employs the same formula embedding architecture. Since NeuroSAT proves *sat* but only *predicts unsat*, we train a classification MLP on top of our trained NeuRes model to further showcase the benefit of our representations on prediction accuracy.

Table 3 confirms this main hypothesis. In essence, this result points to the fact that learning signals obtained from training on *unsat* certificates largely enhance the ability of the neural network to extract useful information from the input formula. This is doubly promising considering NeuroSAT was

Table 3: Performance of full solver mode tested on $\mathbf{SR}(40)$ problems and trained on $\mathbf{SR}(U(10, 40))$ problems where PREDICTED refers to the satisfiability prediction without certificate.

| MODEL | PROVEN (%) | | | PREDICTED (%) | | |
|---|---|---|---|---|---|---|
| | SAT | UNSAT | TOTAL | SAT | UNSAT | TOTAL |
| NEURES | **96.8** | **99.6** | **98.2** | **84.28** | **99.2** | **91.65** |
| NEUROSAT [43] | 70 | - | - | 73 | 96 | 85 |

trained on *millions* of formulas while NeuRes was trained on only *16K* formulas. Lastly, we find that augmenting *sat* formulas by resolution derivations results in relative improvements ($\sim 2.3\%$) in success rate even though these derivations are attempting to prove unsatisfiability.

## 8 Utilizing Model Fan-Out

In our Full-Attention module, we compute $n^2$ scores and only perform the top-score resolution step. This greedy approach arguably underutilizes the attention grid computations as it ignores other high-scoring steps that might lead to a shorter proof thereby improving the success rate in addition to reducing the number of queries to the model. The latter leads to an overall runtime reduction since performing an extra symbolic resolution step is much faster than a forward model pass. As such, we experiment with performing the top $k$ steps of the attention grid after each forward pass. It should be noted, however, that this yields diminishing returns as it leads to a faster growth of the clause base which in turn inflates the attention grid. For $k > 1$, after deriving the empty clause, only clauses that connect to it in the resolution graph are kept in the final proof. This post-processing step is linear in the proof length and eliminates redundant resolution steps resulting from the higher fan-out. Table 4 shows that taking the top 3 steps already yields a massive reduction in proof lengths along with a significant boost to the success rate.

Table 4: Performance of different model fan-outs on $\mathbf{SR}(40)$ test data. Proof length (p-Len) and #Model Calls are both normalized by the length of the teacher proof.

| FULL-ATTN FAN-OUT | P-LEN | #MODEL CALLS | TOTAL PROVEN (%) |
|---|---|---|---|
| TOP-1 | 1.15 | 1.15 | 98.2 |
| TOP-3 | 0.57 | 0.49 | 99.9 |
| TOP-5 | **0.52** | **0.43** | **100.0** |

One way to offset the attention grid inflation with higher fan-out would be to keep a saliency map for all clauses then discarding $k$ clauses with the least saliency scores after each forward pass. One simple way to compute this saliency score for a clause would be the sum/mean/max of its respective row in the attention scores grid. Another proxy for saliency could be the recency score reflecting how many steps have elapsed since the last time a given clause was used.

## 9 Generalizing to Larger Problems

In order to test our model's out-of-distribution performance, we evaluate our NeuRes model on five datasets comprising formulas with up to 5 times more variables than encountered during training. We use the same distributions reported by NeuroSAT and we run our model for the same maximum number of iterations (1000).

Figure 5 shows the scalability of NeuRes to larger problems by letting it run for more iterations. Compared to NeuroSAT [43], NeuRes scores a much higher first-try success rate on all 5 problem distributions, and a higher final success rate on all of them except for $\mathbf{SR}(40)$ on which both models nearly score $100\%$. Particularly, NeuRes shows higher first-try success on the 3 largest problem sizes where NeuroSAT solves zero or near-zero problems on the first try.

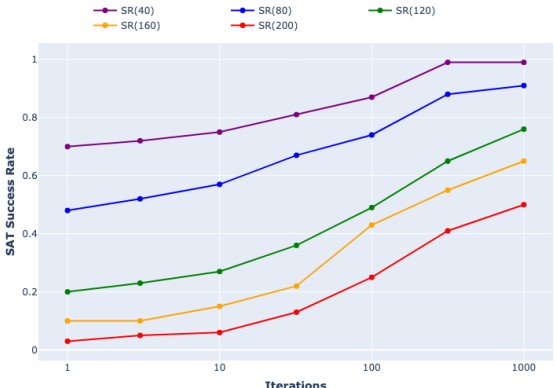

Figure 5: SAT success rate over iterations.

## 10 Conclusion

In this paper, we introduced a deep learning approach for proving and predicting propositional satisfiability. We proposed an architecture that combines graph neural networks with attention mechanisms to generate resolution proofs of unsatisfiability. Unlike methods that merely predict unsatisfiability, our models provide easily verifiable certificates for their verdicts. We demonstrated that our certificate-based training and resolution-aided mode of operation surpass previous approaches in terms of performance and data efficiency, which we attribute to learning better representations.

Despite its promising benchmark performance, our model cannot solely outperform highly engineered industrial solvers, as is currently the case for all neural methods as standalone tools. The gap between neural networks and symbolic algorithms is still rather large, and our hope is to bring deep learning methods one concrete step closer to filling this gap. For NeuRes, this step is recognizing the immense value of carefully integrating certificates into the model design and training as opposed to using shallow supervision labels. Last but not least, it is worth noting that even at their present state, neural networks stand great potential to advance traditional solvers by combining them into hybrid solvers that utilize the deep long-range dependencies captured by neural networks along with the exploration speed of symbolic algorithms. Moreover, we demonstrated a unique potential to advance SAT solving through proof reduction, as proof size is a major challenge in certifying the results of traditional solvers. This proof reduction is facilitated by a bootstrapped training procedure that uses teacher proofs as a guide as opposed to a golden standard.

### Acknowledgments and Disclosure of Funding

This work was supported by the European Research Council (ERC) Grant HYPER (No. 101055412).

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

# Appendix

## A  NeuroSAT Formula Graph Construction

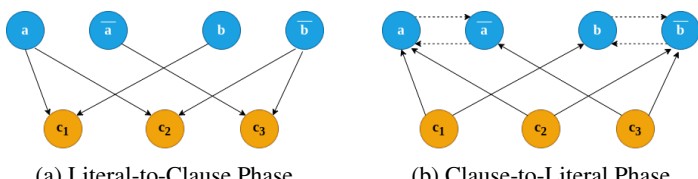

(a) Literal-to-Clause Phase          (b) Clause-to-Literal Phase

Figure 6: Two-phase message-passing round on NeuroSAT formula graph.

NeuroSAT-style formula graphs have two designated node types: clause nodes connected to the literal nodes corresponding to their constituent literals [43]. For example, in Figure 6, the clause contents are as follows: $c_1 = (a \vee b)$, $c_2 = (a \vee \bar{b})$, $c_3 = (\bar{a} \vee b)$. Each message-passing round involves two exchange phases: (1) Literal-to-Clause, and (2) Clause-to-Literal (and implicitly Literal-to-Complement). This construction is particularly efficient as it allows the message-passing protocol to cover the entire graph connectivity in at most $|V| + 1$ rounds where $V$ is the set of variables in the formula.

## B  Clause Connectivity Under Static Embeddings

In Section 4.2, we stated that under static embeddings for a derived clause, as the embedder creates its embedding, it only updates the representations of the variables involved in it – leaving other clause embeddings intact. This might present a problem for Full-Attn where the attention grid contains all clauses including disconnected[1] pairs. An example of such a pair would be two derived clauses that do not share a variable. This could potentially lower the efficacy of the Full-Attn mechanism as it tries to match clauses that are unaware of each other. Interestingly, despite being a relaxation on Full-Attn, Anch-Attn has a distinct edge over Full-Attn under static embeddings in form of the following property:

**Lemma B.1.** *Clauses in the variable-anchored attention grid of Anch-Attn are guaranteed to be connected under both static and dynamic embeddings.*

*Proof.* Let $v$ be a variable in the input formula, and the set of clauses of a $v$-anchored attention grid be $A$. We show that we always have at least one clause $A_i \in A$ that reaches all other clauses in $A$ on the formula graph. We make two case distinctions:

**Case 1:** All clauses in $A$ are input clauses (in the original formula). Here, the lemma follows trivially since all these clause were connected during the input-phase message-passing protocol as they share at least one variable $v$.

**Case 2:** $A$ contains derived clauses. Let $A_i$ be the most recently derived clause in $A$. Since $A_i$ shares variable $v$ with all other clauses in $A$, then $A_i$ would be connected to them all during the derivation-phase message-passing protocol immediately after $A_i$ was derived. This is because $A_i$ receives a message from $V$ (under both static and dynamic embeddings) containing information about all other clauses containing $v$, which is precisely $A \setminus \{A_i\}$. Therefore, the lemma holds.  □

---

[1] We use the terms ***connected*** and ***disconnected*** here to refer to the fact of whether two nodes have exchanged messages (in either direction) or not, respectively.

## C    Teacher Proof Reduction

One rather interesting observation on Table 5 is that the model appears to be marginally better at producing shorter proofs for unseen (test) formulas than for training formulas. While we would normally expect the opposite, a fair speculation would be that the trained model was teacher-forced to match teacher proofs during training over multiple epochs while the same does not hold for unseen formulas where the bias towards teacher behavior is significantly lower. Definitively confirming this would require a more in-depth investigation.

Table 5: Teacher proof reduction statistics of non-bootstrapped model trained on unreduced $\mathbf{SR}(U(10, 40))$ dataset. Note that all rows, except for Total Reduction, are computed over the *reduced portion* of the dataset, i.e., the proofs that were successfully shortened by NeuRes.

| (%) | TRAIN | TEST |
|---|---|---|
| PROOFS REDUCED | 17.82 | 18.29 |
| MAX. REDUCTION | 86.11 | 76.4 |
| AVG. REDUCTION | 23.55 | 23.65 |
| TOTAL REDUCTION | 3.07 | 3.15 |

## D    Runtime Comparison with Traditional Solver

In the following table, we compare the average runtimes of our top-1 Full-Attn, top-3 Full-Attn, and the traditional solver we used as a teacher (BooleForce) on our main $\mathbf{SR}(40)$ test dataset. For both Full-Attn models, we use our Python prototype implementation; for BooleForce, we use an official C implementation.

Table 6: Average time (ms) to solve an instance by neural model vs. teacher solver.

| SOLVER | SAT (MS) | UNSAT (MS) | TOTAL (MS) |
|---|---|---|---|
| FULL-ATTN TOP-1 | 2.3 | 88 | 45.15 |
| FULL-ATTN TOP-3 | 3 | 54.4 | 28.7 |
| BOOLEFORCE | 4 | 5 | 4.5 |

## E    Model Size Comparison with NeuroSAT

In terms of the model architecture, both NeuRes and NeuroSAT models can be broken down to:

- **Embedding/Representation Network:** for both models, this network is an LSTM-based GNN that embeds the formula graph by message-passing. We use the exact same architecture and model size to ensure that our improved representations are a result of our fully certificate-based learning objective as opposed to a tweak in the model architecture. This GNN has $429,824$ parameters in total.

- **Downstream Networks:** NeuroSAT: uses a 3-layer MLP applied on the literal embeddings (width $= 128$) to extract the literal votes to predict if the formula is satisfiable or not. This MLP has $128 \times 128 \times 3 = 49,152$ parameters. NeuRes (Full-Attn): uses an attention module to select clause pairs. This attention network is composed of two 1-layer MLPs for the query and key transformations on the clauses embeddings (width $= 128$). The whole attention module has $128 \times 128 \times 2 = 32,768$ parameters. To decode the variable assignments, NeuRes uses a 2-layer scalar MLP with $128 \times 128 + 128 = 16,512$ parameters

Total NeuroSAT size $= 429,824 + 49,152 = 478,976$ parameters

Total NeuRes size $= 429,824 + 49,280 = 479,104$ parameters

All in all, NeuRes only learns 128 more parameters.

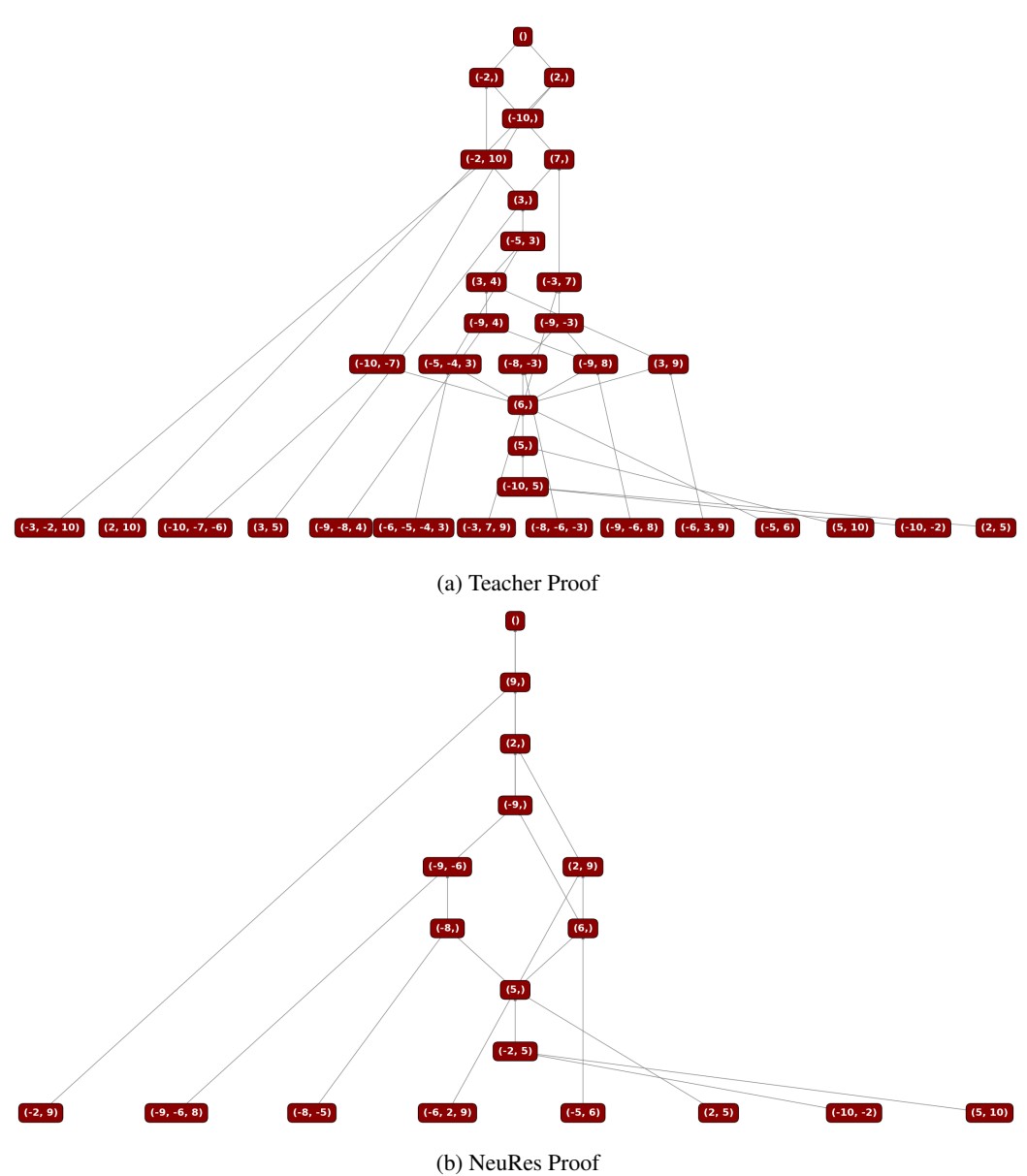

(a) Teacher Proof

(b) NeuRes Proof

Figure 7: Teacher Proof Reduction Example

