# OpenReview forum: "Learning Better Representations From Less Data For Propositional Satisfiability"
_NeurIPS.cc/2024/Conference — NeurIPS 2024 spotlight_

### Official Review · Reviewer_hPQb · 2024-07-09

**Soundness:** 3
**Presentation:** 3
**Contribution:** 3
**Rating:** 6
**Confidence:** 2

**Summary:**

The paper presents NeuRes, a neural symbolic system designed to solve the Boolean satisfiability problem of CNF (Conjunctive Normal Form). NeuRes utilizes a message-passing graph neural network to embed the information contained in CNF and attempts to predict clause pairs for propositional resolution and value assignments for satisfiability. NeuRes proposes three novel attention mechanisms to better select clause pairs for resolution and employs expert iteration to progressively optimize the generated proofs. The experimental results demonstrate the effectiveness of the proposed NeuRes system.

**Strengths:**

- The approach proposed in the paper is novel, featuring a parallel mechanism for concurrently proving unsatisfiability and attempting to find a truth value assignment. This mechanism is expected to speed up the proving process significantly.
- The three proposed attention variants each have distinct characteristics and are suitable for different situations. The expert iteration process effectively shortens the proof length.
- The paper is well-written and easy to follow.

**Weaknesses:**

- The paper does not include a thorough ablation study to evaluate the effectiveness of each proposed component. i.e. It lacks an assessment of the true value assignment task’s effectiveness.
- As noted in the Limitation (Section 9), the efficiency of the neural method is concerning. It adds substantial computational overhead for clause selection using the neural network. However, overall, I believe this work represents a solid advancement in neuro-symbolic approaches for SAT solvers.

**Questions:**

- Is there a comparison of the efficiency of the three different attention mechanisms as well as the baseline method (NeuroSAT)? It appears that full attention produces the most promising results, but it is also the least efficient.

**Limitations:**

Yes, the authors adequately addressed the limitations and posed no negative societal impact.

---

> ### Author Rebuttal · Authors · 2024-08-07
>
> Thank you for your review and comments. Please find them addressed below.
>
> > The paper does not include a thorough ablation study to evaluate the effectiveness of each proposed component. i.e. It lacks an assessment of the true value assignment task’s effectiveness.
>
> - It is true that in the paper, we focus on ablations regarding the clause-pair selection mechanisms (i.e., Casc-Attn, Full-Attn, Anch-Attn). The assignment decoder is kept simple on purpose: It is a small MLP, such that we can adequately assess the quality of the learned representations instead of the architecture. This is a quite common approach in the representation probing literature (e.g., \[1\], \[2\], \[3\]). Besides this, we preformed several ablation experiments that we are happy to include in an appendix to the final version of this paper.
>
> > Q1: Is there a comparison of the efficiency of the three different attention mechanisms as well as the baseline method (NeuroSAT)?
>
> - It depends on how you define efficiency. If you refer to the cost of a single forward pass through the model, then for a large enough problem size, Cascaded Attention should be the fastest followed by Anchored Attention then Full Attention. Our attention mechanisms do not directly compare to NeuroSAT since they are used for proving UNSAT which is not done by NeuroSAT (it's one of our contributions).
> - However, a more meaningful metric of efficiency should also take into account the number of steps (fwd passes) a clause selector needs to solve a given formula along with its success rate. As such, one way to compare the different attention mechanisms in terms of efficiency (on a formula distribution of size n) would be according to the following metric:
>   - $\text{Efficiency} = (\text{p-Len} \* \text{Success Rate}) / (\text{time cost of a fwd pass})$
>   - For instance, on the $SR(U(10, 40))$ distribution, the efficiencies of our 3 attention mechanisms would be as follows:
>     - $E(\text{Casc-Attn}) = (1.79 \* 0.3733) / (1.5 ms) = 0.445$
>     - $E(\text{Anch-Attn}) = (2.28 \* 0.605) / (0.6 ms) = 2.299$
>     - $E(\text{Full-Attn}) = (1.67 \* 0.952) / (0.5 ms) = 3.1797$
>   - This means that for that particular distribution, Full-Attn is actually most efficient as the 2D attention grid is faster to compute than making two attention queries (as in Casc-Attn). However, that order of efficiency might vary for a different distribution.
> - For a general runtime breakdown comparison between NeuRes and NeuroSAT:
>   - For **predicting** SAT status, both models have the same runtime as they use the same message-passing GNN to obtain formula embeddings and the same voting MLP to get the SAT prediction.
>   - For **solving** SAT (i.e., with certificate/proof), the runtime depends on the instance size and SAT status as follows:
>     - **SAT**: the formula is solved when a satisfying truth assignment is found. In this case, the runtime is decided by two factors:
>       - #Attempts: Through our experiments, we show that NeuRes finds satisfying assignments to many more SAT instances after 1000 attempts. To see this, you can compare Figure 5 in our paper vs. Figure 5 in the NeuroSAT paper.
>       - The time cost of a single attempt: at any step, NeuRes extracts the full assignment in a single forward pass through a 2-Layer MLP over the literal embeddings. In contrast, NeuroSAT performs a k-means clustering on literal embeddings to separate them into two 2 clusters against a predicate. The latter is significantly more costly as it performs an iterative clustering process that could take hundreds of iterations to converge (the default limit is set to 300 iterations in the NeuroSAT implementation).
>     - **UNSAT**: the formula is solved when the model derives a full (resolution) proof of the empty clause (falsum) from it. Since NeuroSAT does not solve UNSAT formulas, a runtime comparison is not possible (UNSAT cores are not proofs). However, for a general idea, the NeuRes runtime in UNSAT cases depends on the number of resolution steps needed to derive the empty clause. In our evaluation, we've shown that NeuRes in many cases produces much shorter proofs than the teacher traditional solver, and on average to be around 1.15x in length. This is under our 1-step-per-forward-pass model. We thought that this arguably underutilizes our attention grid by only taking the top-1 clause pair, so we performed a side experiment in the rebuttals period to explore potential gains of taking the top 3 clause pairs at each step and perform them. The idea is to reduce the total number of forward passes to the model by performing more resolution steps per fwd pass. On SR(40) test data, this led to a 57.4% reduction in episode lengths (total number of forward passes), model proofs that are on average 43% shorter than the teacher's (p-Len = 0.57 vs the previous 1.15), in addition to a +1.7% total success/proof rate (99.9% vs the previous 98.2%). We intend to include this experiment in the main paper for the sheer performance boost it unlocks.
>
> **References:**
>
> \[1\] Akhondzadeh, Mohammad Sadegh, Vijay Lingam, and Aleksandar Bojchevski. "Probing graph representations." International Conference on Artificial Intelligence and Statistics. PMLR, 2023.
>
> \[2\] Alain, Guillaume, and Yoshua Bengio. "Understanding intermediate layers using linear classifier probes." arXiv preprint arXiv:1610.01644 (2016).
>
> \[3\] Pimentel, Tiago, et al. "Information-theoretic probing for linguistic structure." arXiv preprint arXiv:2004.03061 (2020).

---

### Official Review · Reviewer_tfEq · 2024-07-09

**Soundness:** 3
**Presentation:** 3
**Contribution:** 3
**Rating:** 7
**Confidence:** 3

**Summary:**

The paper integrates attention-based neural networks into the SAT solving process based on resolution. The neural networks aim to predict pairs of disjunctive clauses to merge. The paper proposes several variants of neural networks for this task and conducted experiments to check their performance. Although the resolution is the algorithm to check unsatisfiability, the proposed framework simultaneously addresses the satisfiability task by decoding the embeddings of formulas.

**Strengths:**

+ The address problem, the unsatisfiability of propositional formulas, is a fundamental, important problem in computer science.
+ Unifying resolution, a classic SAT solving algorithm, and attention-based neural networks.
+ The paper considers extensive variants of the neural models and checks their experimental performance.
+ It is experimentally demonstrated that the proposed approach outperforms the previous work, NeuroSat, and the expert iteration can shorten generated unsatisfiability proofs.

**Weaknesses:**

- It is not clear how big the gap between the proposed approach and the highly-engineered SAT solvers is. It is not critical that the highly-engineered SAT solvers work better, but revealing the gap in the current state clarifies the current position of neural SAT solvers.
- Some aspects of the proposed approach and experiment design are unclear (see the below question).

**Questions:**

* How big is the gap between the proposed approach and the highly-engineered SAT solvers?
* What is $\overline{E^L}$? How different is it from $E^L$?
* The advantage of Casc-Attn is claimed to be "it ... can be used to select a tuple of arbitrary length." Does this advantage is utilized in the implementation?
* The paper says that at train time only the positive literal embeddings are used to derive truth assignments, but at test time, the negative literal embeddings are used. Why is this okay although the derivation of truth assignments from the negative embeddings should not be learned?
* What is the ratio between sat and unsat problems in the synthesized dataset?
* At the beginning of Section 6, the paper mentions the training and test datasets, but does not the validation dataset. I think it should be used because hyperparameters should be adjusted. What validation dataset is used?
* (Minor) What is Res-Aided model (mentioned at the beginning of Section 8)? I find this terminology only there.

**Limitations:**

I think the limitations are discussed appropriately.

---

> ### Author Rebuttal · Authors · 2024-08-07
>
> Thank you for your feedback and questions. Please find our responses to your questions below.
>
> > Q1: How big is the gap between the proposed approach and the highly-engineered SAT solvers?
>
> - Generally speaking, the main merit of our approach over traditional SAT solvers is its ability to capture deep insights into the underlying data distribution, which for instance, enables it to come up with far shorter proofs than the traditional solver with no extra supervision than the initial imperfect ground truth it. This is both an advantage over traditional solvers and traditional neural approaches that are limited by the quality of their initial labels. We mainly owe that to our sound-by-design resolution module that enables the network to safely (i.e., soundly) explore other viable solutions without distorting the semantics of the problem. That being said, these deep insights come at a cost of heavier computations involved in neural networks compared to traditional heuristic solvers.  For instance, on SR(40) data, it takes NeuRes an average of \~45 ms (2.3 ms for sat and 88 ms for unsat) to solve an instance in our Python implementation while it takes \~4 ms to solve it on a C implementation of the BooleForce solver. Different machinery (GPU for neural networks vs CPU for traditional solvers) make this comparison further complicated.
>
> > Q2: What is $\overline{E^L}$? How different is it from $E^L$?
>
> - $\overline{E^L}$ is mean of $E^L$ (i.e., mean literal embedding). $E^L$ is the matrix containing all literal embeddings of dimensions $m$x$d$ while $\overline{E^L}$ is a vector of width $d$, where $m$ is the number of literals and $d$ is the embedding width.
>
> > Q3: The advantage of Casc-Attn is claimed to be "it ... can be used to select a tuple of arbitrary length." Does this advantage is utilized in the implementation?
>
> - We do not utilize this advantage in our implementation to get a direct comparison with the other two variants that only perform binary resolution. That being said, generalized resolution steps (involving a tuple of clauses) can always be easily broken into a sequence of binary resolution steps (involving pairs of clauses). However, we have not performed that experiment.
>
> > Q4: ...at train time only the positive literal embeddings are used to derive truth assignments, but at test time, the negative literal embeddings are used. Why is this okay..?
>
> - We previously performed several experiments in that direction where we tried supervising both positive and negative literals (and combining both losses with different aggregates: sum, min, & max) against the ground truth assignment. This surprisingly did not lead to an improvement compared to only supervising positive literals. At test time, however, we found using negative literals (in addition to the positive ones) sometimes produced satisfying assignments earlier. We do not have a precise account for this observation, but our best intuition attributed this result to the fact that the formula graph has no explicit notion of +ve and -ve literals per se, it's just a matter of which literal is connected/adjacent to which clauses (+ve and -ve literals are connected by a special edge). As such, both literal nodes have a different local view into the rest of formula, which could result in one of them leading to a satisfying assignment faster than the other.
>
> > Q5: What is the ratio between sat and unsat problems in the synthesized dataset?
>
> - It's 50-50% across all training, validation, and test datasets to eliminate bias towards one class over the other.
>
> > Q6: What validation dataset is used?
>
> - For the UNSAT-only experiments (Section 6), we use a validation dataset containing 1K UNSAT formulas of the $SR(U(10, 40))$ distribution. For the full solver experiments (Section 7), we use a validation set of 2K formulas (1K SAT + 1K UNSAT) of the same distribution.
>
> > Q7: What is Res-Aided model?
>
> - The Res-Aided (i.e., Resolution-Aided) model refers to the full solver (SAT + UNSAT). The name is based on the rationale that the resolution head aids the assignment search for satisfiable formulas. (Res is usually used to refer to the resolution operator in the literature).

---

> > ### Comment · Reviewer_tfEq · 2024-08-09
> >
> > Thank you for the response. I just hope my concerns are clarified also in the revision if possible.

---

> > > ### Author Response · Authors · 2024-08-11
> > >
> > > Thank you again for your valuable feedback. We shall include clarifications on the highlights you mentioned mainly in form of a passage on the gap between neural and traditional solvers along with the full ablation experiment of literal assignment supervision.

---

### Official Review · Reviewer_64Vk · 2024-07-13

**Soundness:** 3
**Presentation:** 3
**Contribution:** 3
**Rating:** 6
**Confidence:** 2

**Summary:**

The paper presents NeuRes, a system to solve SAT/UNSAT problems. In particular, it is able to provide UNSAT certificates via resolution proofs. The system constructs a resolution proof by iteratively selecting two clauses from all clauses, producing the resolvent from the two clauses, and adding it back to the clause formula. The selection of two clauses is done by a selector network. They experiment with different attention mechanisms in the selector network, including full attention, selecting an anchor variable and then doing full attention on the subset of clauses including the anchor variable, etc. When training, they use an off-the-shelf symbolic solver to generate resolution proofs and train to predict the clause selection with a teacher forcing style. The experiments show that NeuRes is able to generate resolution proofs for UNSAT problems (99.6% problems solved) for small problems with 40 variables, and also shows some generalizability to larger problems.

**Strengths:**

* The method is able to generate resolution proofs for UNSAT problems, which is a certificate for UNSAT problems, unlike previous methods such as NeuroSAT.
* The method shows some generalizability to larger problems, which is a good sign for the method to be applied to larger problems.
* The method is sometimes able to generate shorter resolution proofs than the teacher, and they propose a bootstrapping method to further improve the performance based on those shorter proofs.

**Weaknesses:**

* Resolution-based UNSAT certification may not be as useful or practical as clausal proofs like the DRAT format, due to resolution-based proofs potentially being too large for large SAT instances.

**Questions:**

* Could you explain how the method applies to SAT problems? If it's an UNSAT problem, then I can see that it terminates at deriving an empty clause, but how does it work for SAT problems?
* How does the model size of NeuRes compare to the baseline NeuroSAT on the SAT problems? How does the solving time compare between the two methods?

**Limitations:**

The limitation about the performance of neural methods on SAT solving still lags behind symbolic solvers is discussed.

---

> ### Author Rebuttal · Authors · 2024-08-07
>
> Thank you for your review and comments. Please find your questions addressed below.
>
> > Q1: Could you explain how the method applies to SAT problems?
>
> - In the case of a satisfiable formula, NeuRes terminates when a satisfying truth assignment is found by the assignment decoder network. Initially the model doesn't know the SAT status of the formula, so at each step, it performs two operations: 1) derives a new clause by resolution, and 2) produces a candidate truth assignment. Track 1 guides the assignment search by making new inferences (clauses) on the formula. As you mentioned, track (1) proves UNSAT upon deriving the empty clause while track 2 terminates upon finding an assignment that satisfies all clauses.
>
> > Q2.1: How does the model size of NeuRes compare to the baseline NeuroSAT on the SAT problems?
>
> - In terms of the model architecture, both models can be broken down to:
>   - **Embedding/Representation Network**: for both models, this network is an LSTM-based GNN that embeds the formula graph by message-passing. We use the exact same architecture and model size to ensure that our improved representations are a result of our fully certificate-based learning objective as opposed to a tweak in the model architecture. This GNN has 429,824 parameters in total.
>   - **Downstream Networks**:
>     - NeuroSAT: uses a 3-layer MLP applied on the literal embeddings (width=128) to extract the literal votes to predict if the formula is satisfiable or not. This MLP has 128\*128\*3 = 49,152 parameters.
>     - NeuRes (Full-Attn): uses an attention module to select clause pairs. This attention network is composed of two 1-layer MLPs for the query and key transformations on the clauses embeddings (width=128). The whole attention module has 128\*128\*2 = 32,768 parameters. To decode the variable assignments, NeuRes uses a 2-layer scalar MLP with 128\*128 + 128 = 16,512 parameters
>
>
>
> Total NeuroSAT size = 429,824 + 49,152 = 478,976 parameters
> Total NeuRes size = 429,824 + 49,280 = 479,104 parameters
>
> (i.e., NeuRes only learns 128 more parameters)
>
> > Q2.2: How does the solving time compare between the two methods?
>
> - For **predicting** SAT status, both models have the same runtime as they use the same message-passing GNN to obtain formula embeddings and the same voting MLP to get the SAT prediction.
> - For **solving** SAT (i.e., with certificate/proof), the runtime depends on the instance size and SAT status as follows:
>   - **SAT**: the formula is solved when a satisfying truth assignment is found. In this case, the runtime is decided by two factors:
>     - #Attempts: Through our experiments, we show that NeuRes finds satisfying assignments to many more SAT instances after 1000 attempts. To see this, you can compare Figure 5 in our paper vs. Figure 5 in the NeuroSAT paper.
>     - The time cost of a single attempt: at any step, NeuRes extracts the full assignment in a single forward pass through a 2-Layer MLP over the literal embeddings. In contrast, NeuroSAT performs a k-means clustering on literal embeddings to separate them into two 2 clusters against a predicate. The latter is significantly more costly as it performs an iterative clustering process that could take hundreds of iterations to converge (the default limit is set to 300 iterations in the NeuroSAT implementation).
>   - **UNSAT**: the formula is solved when the model derives a full (resolution) proof of the empty clause (falsum) from it. Since NeuroSAT does not solve UNSAT formulas, a runtime comparison is not possible (UNSAT cores are not proofs). However, for a general idea, the NeuRes runtime in UNSAT cases depends on the number of resolution steps needed to derive the empty clause. In our evaluation, we've shown that NeuRes in many cases produces much shorter proofs than the teacher traditional solver, and on average to be around 1.15x in length. This is under our 1-step-per-forward-pass model. We thought that this arguably underutilizes our attention grid by only taking the top-1 clause pair, so we performed a side experiment in the rebuttals period to explore potential gains of taking the top 3 clause pairs at each step and perform them. The idea is to reduce the total number of forward passes to the model by performing more resolution steps per fwd pass. On SR(40) test data, this led to a 57.4% reduction in episode lengths (total number of forward passes), model proofs that are on average 43% shorter than the teacher's (p-Len = 0.57 vs the previous 1.15), in addition to a +1.7% total success/proof rate (99.9% vs the previous 98.2%). We will include this experiment in the main paper for the sheer performance boost it unlocked.

---

### Official Review · Reviewer_R9jd · 2024-07-13

**Soundness:** 3
**Presentation:** 4
**Contribution:** 3
**Rating:** 8
**Confidence:** 4

**Summary:**

The authors proposed an attention-based architecture that autoregressively selects pairs of clauses for propositional resolution. The framework can generate proofs of unsatisfiability and accelerate the process of finding satisfying truth assignments simutaneously. Emprical evaluation shows that the resulting model achieves better performance than NeuroSAT in terms of both correctly classified and proven instances.

**Strengths:**

+ The model learns to pick resolution pairs rather than naively predicts binary satisfiability, resulting in an overall elegant approach that combines classic symbolic reasoning and machine learning. An unsat "prediction" always comes with a valid proof, and a sat prediction is alway correct.
+ Detailed attention mechanisms are presented in a progressive manner. Each attention mechanism is accompanied by proper motivation and explanation. The approach is clearly well thought-out and optimized carefully. Again, not applying pretrained language models blatantly stands out in the flooded submissions today.
+ I like the idea of predicting additionally the truth assignment. Although in principle the assignment can be read off once the resolution is complete, it's always better to terminate early. And by learning to pick the resolution pairs and assigning truth values simultaneously, more semantics can hopefully be injected into the models. Well done!

**Weaknesses:**

- Speed can potentially be an issue since every resolution step invokes a forward pass of the model, and for large problems, the calls can be quite numerous.

**Questions:**

- How easy do you think the approach can be combined with traditional SAT optimization?
- Have you thought about extending the approach to anything beyond propositional SAT?

**Limitations:**

The authors discussed the limitation in section 9 and I agree with them.

---

> ### Author Rebuttal · Authors · 2024-08-07
>
> Thank you for your feedback and questions; we're glad you enjoyed the paper!
>
> We address your comments in the following.
>
> > Speed can potentially be an issue.. for large problems, the calls can be quite numerous.
>
> - It is true that speed is a limiting factor in scaling deep learning models to very large instances. Particularly, our model inherits the scaling laws of transformers (due to self-attention) which famously limits LLMs' ability to handle very long sequences. In our context, there are two ways to tackle this:
>
>   **(1) Reduce the cost of a single forward pass**: this can be done in numerous ways, but we want to highlight a particularly relevant one of reducing the size of the attention grid. We already showed an example of that in Anchored-Attention, but there are generally more ways to only compute attention scores on a subset of clauses. This subset of interest can either be determined by a heuristic or by a learned scoring model (e.g., MLP). Keeping a constant-sized attention window would drastically alleviate scaling constraints on our network at a certain expense of limiting exploration.
>
>   **(2) Reduce the total number of calls**: in our Full-Attention module, we compute $N^2$ scores and only take the top-score resolution step. This arguably underutilizes the attention grid. On that basis, we conducted a small ablation (over the course of these rebuttals) where we perform the top 3 resolutions instead of only the top 1. This led to notable gains across all our main metrics. We report our findings on $SR(40)$ test data in the table below. Proof length and #Model Calls are both normalized by the length of the teacher proof.
>
> | Variant | Proof Length (normalized) | #Model Calls (normalized) | Total Proven (%) |
> |---------|---------------------------|---------------------------|------------------|
> | Top-1   | 1.15                      | 1.15                      | 98.2             |
> | Top-3   | **0.57**                      | **0.49**                     | **99.9**             |
>
> > Q1: How easy do you think the approach can be combined with traditional SAT optimization?
>
> - One promising way to combine our approach with traditional CDCL-style SAT solvers is to use our models as variable selection heuristics (i.e., to decide which variable to branch on). NeuroCore \[5\] trained a simplified NeuroSAT model to assign scores to variables reflecting how likely they are to cause a conflict, which led to a significant boost to a traditional MiniSat solver where it solved 10% more problems within standard timeout on SATCOMP-2018. Since our approach produces better representations over NeuroSAT for SAT and UNSAT prediction, this suggests a natural way to combine and improve traditional solvers with our approach. It is worth mentioning that gathering training data for SATCOMP was a major challenge for the NeuroCore paper, so the data efficiency of NeuRes would be expected to pay off in that context.
> - Another way to create a hybrid solver would be periodically sampling resolution steps from NeuRes to augment a traditional CDCL solver. The rationale here is that the full resolution proof can be too costly, but intermediate resolution derivations could help find conflicts earlier during the branching which helps the solver save exploration time.
>
> > Q2: Have you thought about extending the approach to anything beyond propositional SAT?
>
> - At the moment, we are mainly interested in extending our approach to Quantified Boolean Formulas (QBF), for which a sound and complete resolution calculus exists \[1\]. QBF allows succinct expression of problems arising in AI planning \[2\] and verification \[3\], but is significantly less studied than propositional SAT.
> - QBF is a somewhat straightforward extension to NeuRes as it only requires the following modifications:
>   - Learning an extra variable embedding initialization: since in QBF, variables are either universally or existentially quantified, different initial embeddings should be learned to set both types of variables apart prior to message-passing.
>   - Encode quantification ordering over the variables to reflect the difference between $\\exists x \\: \\forall y$ and $\\forall y \\: \\exists x$. This can be done using positional encoding for instance.
> - In the long term, we are also interested in generalizing the insights from our resolution-based architectures to theorem proving in general, which is an area that has received significant attention from the machine-learning community. \[4\] gives a good overview.
>
>
> **References:**
>
> \[1\] Hans Kleine Büning, Marek Karpinski, and Andreas Flögel. "Resolution for Quantified Boolean Formulas." Information and Computation 117, 12-18 (1995).
>
> \[2\] Irfansha Shaik and Jaco van de Pol. "Planning as QBF without Grounding." International Conference on Automated Planning and Scheduling, ICAPS 2022.
>
> \[3\] Tzu-Han Hsu, César Sánchez, and Borzoo Bonakdarpour. "Bounded Model Checking for Hyperproperties". International Conference on Tools and Algorithms for the Construction and Analysis of Systems, TACAS 2021.
>
> \[4\] Markus N. Rabe and Christian Szegedy. "Towards the Automatic Mathematician". International Conference on Automated Deduction, CADE 2021.
>
> \[5\] Selsam, Daniel, and Nikolaj Bjørner. "Guiding high-performance SAT solvers with unsat-core predictions." Theory and Applications of Satisfiability Testing–SAT 2019: 22nd International Conference, SAT 2019, Lisbon, Portugal, July 9–12, 2019, Proceedings 22. Springer International Publishing, 2019.

---

> > ### Comment · Reviewer_R9jd · 2024-08-12
> >
> > Thank you for the rebuttal.
> >
> > I am happy to see that all my concerns are addressed. One more thing, can you please tell me how much time (in seconds) it took on average on the problems you tested? A rough estimate would be fine, and I won’t hold it against you even if it’s slower compared to traditional methods.

---

> > > ### Author Response · Authors · 2024-08-13
> > >
> > > We would like to thank the reviewer again and are happy that all their concerns have been addressed.
> > >
> > > In the meantime, we performed a time profiling for our top-3 model to confirm that it shortens the runtime as well as the proof/episode lengths. In the following table, we compare the average runtimes of our top-1 Full-Attn, top-3 Full-Attn, and the traditional solver we used as a teacher (BooleForce) on our main SR(40) test dataset. For both Full-Attn models, we use our Python prototype implementation; for BooleForce, we use an official C implementation.
> > >
> > > | Solver          | SAT (ms) | UNSAT (ms) | Combined (ms) |
> > > |-----------------|----------|------------|---------------|
> > > | Top-1 Full-Attn | 2.3      | 88         | 45.15         |
> > > | Top-3 Full-Attn | 3        | 54.4       | 28.7          |
> > > | BooleForce      | 4        | 5          | 4.5           |

---

> > > > ### Comment · Reviewer_R9jd · 2024-08-13
> > > >
> > > > Thanks for the response.
> > > >
> > > > After reading the discussion between the authors and reviewer MVHU, I am inclined to believe that the authors did not misrepresent their contribution. I agree with reviewer MVHU that directly comparing the SAT solving capability of NeuroSAT and NeuRes isn’t entirely fair because NeuRes is built on resolution, which is by itself complete.
> > > >
> > > > That said, the authors made it clear in the title that the primary focus is on learning a better representation. This is achieved by using the same embedding architecture but learning from a different objective than NeuroSAT. In this context, I believe the title fits the paper very well, and perhaps it is the resulting resolution prover with learned heuristics that should be considered a byproduct here.
> > > >
> > > > Overall, the authors helped me understand their contribution better through the rebuttal. I would like to see this paper occur in this year’s NeurIPS, so I am raising my score to a strong accept.

---

### Official Review · Reviewer_MVHU · 2024-07-16

**Soundness:** 3
**Presentation:** 1
**Contribution:** 3
**Rating:** 4
**Confidence:** 4

**Summary:**

This paper presents a deep-learning-based approach for generating resolution proofs for SAT formulas. The proposed method outperforms NeuroSAT on proving/predicting satisfiability on a family of benchmarks.

**Strengths:**

- Instead of directly learning to predict satisfiability, this paper proposes to learn the strategy to apply the resolution rule. I like this general approach, as it can provide actual proofs of both satisfiability and unsatisfiability.
- Breaking down the resolution strategy prediction task to two steps (i.e., choosing a variable and then choosing two clauses) is a nice optimization that reduces the computational cost.
- The approaches are explained clearly and the paper is relatively self-contained.

**Weaknesses:**

- At its core, the key technical contribution of the paper is that instead of learning an end-to-end SAT solver, the paper proposes to learn a resolution strategy that selects the clauses to resolve for a given formula. This is a fine idea and similar in spirit to many existing lines of work in ML applied to constraint solving, where the goal is to learn a particular heuristics (e.g., branching, restart) to replace some hand-crafted ones. However, to evaluate the proposed method, the paper should compare with existing resolution strategies (potentially including learning-based ones if they exist already) on statistics such as proof lengths and runtime. The comparison with NeuroSAT to me is quite apple-to-orange.
 - Related to the point above, the paper is entitled "learning better representations from less data", instead of something along the line of "learning to perform resolution". The current title can be a little misleading because words like "better" and "less" do not make much sense when the underlying learning task changes (from satisfiability prediction to resolution clause selection).
- The paper only considers very simple SAT instances and it seems difficult to scale the presented approach to much larger real-world instances.

**Questions:**

1. Could you compare the learned resolution strategy with existing ones?
2. Have you conducted ablation studies of the effect of the model architecture to proof length/success rate?

**Limitations:**

The paper discusses the scalability/efficiency limitation of the proposed method. I do not see the potential negative societal impact of their work.

---

> ### Author Rebuttal · Authors · 2024-08-07
>
> We thank the reviewer for their comments, and we address their reservations below.
>
> > The comparison with NeuroSAT to me is quite apple-to-orange.
>
> > The current title can be a little misleading because words like "better" and "less" do not make much sense when the underlying learning task changes (from satisfiability prediction to resolution clause selection).
>
> - The learning objectives indeed differ, but the general task remains the same, i.e., propositional SAT. The difference is that our approach (more ideally) decides the problem by proof (resolution or assignment) not just by prediction, which cannot be easily verified. Reframing the learning objective is a major aspect of innovation in machine learning methods; in our case, we reframe it from prediction to certificate generation. Note that we also compare with NeuroSAT in terms of **prediction accuracy** and **assignment decoding success rate**, both of which are 1-to-1 comparisons.
> - "Better" and "less" in our context refer to two specific outcomes of our project:
>   - We claim better representations based on the fact that the same prediction network achieves notably higher accuracy when trained on our representations vs. the NeuroSAT representations.
>   - "less data" here directly refers to the fact that our model is trained on two orders of magnitude fewer data samples (formulas) than NeuroSAT (**16K** vs. **several millions**) and on the same data distribution $SR(U(10, 40))$.
>
> > Q1: Could you compare the learned resolution strategy with existing ones?
>
> - To the best of our knowledge, NeuRes is the first learning-based approach for propositional resolution in the literature.
> - The $\\text{p-Len}$ metric reflects a comparison with the BooleForce solver (used as a teacher) in terms of proof length. To get a numeric sense about the runtime, on $SR(40)$ formulas, it takes NeuRes an average of \~45 ms (2.3 ms for sat and 88 ms for unsat) to solve an instance in our Python prototype implementation while it takes \~4 ms to solve it on a C implementation of the BooleForce solver. Although NeuRes produces much shorter proofs than BooleForce in many cases, it still lags behind it in terms of pure runtime, as is generally the case for neural methods at the current state of the art.
> - We do not compare with more traditional solvers (other than BooleForce) since the objective of our paper is more towards improving the learning aspects of neural solvers as opposed to competing with traditional symbolic solvers.
>
> > Q2: Have you conducted ablation studies of the effect of the model architecture to proof length/success rate?
>
> Over the course of this project, we conducted numerous experiments that we intend to include in an appendix because they were not essential to the main findings except for the three main **attention mechanisms/architectures** (which we already ablate in the main paper). We are happy to include these in an appendix to the paper. Please find some of our other ablations below:
>
> **Further ablations:**
>
> - Important Notes:
>   - We decided to not modify the GNN architecture to ensure our improved performance is not a result of a mere architecture tweak on NeuroSAT by adopting the same network and mechanism to obtain the initial representations.
>   - Our downstream networks (assignment decoder and prediction head) are both simple and small MLPs to clearly highlight the impact of our certificate-driven learning setting/objective. This is a quite common approach towards assessing the quality of learned representation by how well a simple downstream model performs on them. Examples of this can be found in the representation probing literature \[1\] \[2\].
> - **Global Context Vector**: We experimented with using an LSTM **encoder-decoder network** that takes in the initial clause embeddings and produces a global summary/context vector (its last hidden state) to condition our downstream networks on. This was in fact our original architecture, but we later found out that it was unnecessary to (and even slightly hurts) our model performance since our dynamic embedding already captures the relevant state information locally in the clause embedding, and so conditioning our attention module on the global context vector did not add new information. We report the performance comparison between this variant and our simplified  model (w/o the encoder-decoder) on SR(40) test data below (total over both SAT and UNSAT):
>
>   | Variant                    | Proven (%) | Predicted (%) |
>   |----------------------------|------------|---------------|
>   | Global Context (Enc-Dec)   | 97.4       | 91.35         |
>   | Local Context (simplified) | **98.2**       | **91.65**         |
> - **Clause Pair Scoring (Full-Attn)**: the resolution operator is symmetric on clause order while attention is not. That is, for each clause pair $(c_1, c_2)$, their attention score on the upper triangle of the attention grid $S(c_1, c_2)$ is not necessarily the same as on the lower triangle $S(c_2, c_1)$. We experimented with different ways to combine both scores, and found that only taking the upper triangle score performs best (and is also faster).
>
> | Scoring Scheme    | Proven (%) | Predicted (%) |
> |-------------------|------------|---------------|
> | Upper Triangle    | **98.2**       | **91.65**         |
> | Upper + Lower     | 97.25      | 91.35         |
> | Max(Upper, Lower) | 97.5       | 91.4          |
> | Min(Upper, Lower) | 51.1       | 86.15         |
>
> **References:**
>
> \[1\] Akhondzadeh, Mohammad Sadegh, Vijay Lingam, and Aleksandar Bojchevski. "Probing graph representations." International Conference on Artificial Intelligence and Statistics. PMLR, 2023.
>
> \[2\] Alain, Guillaume, and Yoshua Bengio. "Understanding intermediate layers using linear classifier probes." arXiv preprint arXiv:1610.01644 (2016).

---

> ### Comment · Reviewer_MVHU · 2024-08-12
>
> I thank the authors for their answers, which help me understand their perspective a little better. I would also like to re-emphasize that I like the general methodology this paper is taking. However, I still think that the paper has some serious presentation issues.
>
> To me, NeuRes is better categorized as an ML-based resolution heuristics to be installed in a well-established SAT-solving algorithm, i.e., the resolution-based *Davis-Putnam (DP) algorithm*.  This is fundamentally different from NeuroSAT, which is tackling a different research question: to what extend can we learn a SAT-solving procedure completely from scratch? For this reason, it is quite odd that a large portion of the experimental evaluation is focused on comparison against NeuroSAT and the paper seems to position itself as an improved solution over NeuroSAT, while the two methods are **incomparable**.
>
> Consider the comparison on **Proven %** in Table 3. The configuration NeuRes is essentially running a DP-like algorithm (i.e., iteratively performing resolution) while repeatedly using a neural network as the heuristic to decide which variable and clauses to choose. This algorithm is guaranteed to generate a proof, *regardless of the heuristic*. On the other hand, the configuration NeuroSAT performs one inference of the neural network to generate a proof (of satisfiability). The fact that NeuRes generates a correct proof more frequently than NeuroSAT is not surprising and does not say much about NeuRes's effectiveness, because given a sufficient number of resolution steps, *any* resolution heuristic can generate a correct proof 100% of the time. The paper does try to make the comparison "fairer" by putting a upper bound on the number of resolution steps that the configuration NeuRes can perform in Table 3. However, this upper bound is **very loose** (e.g., 4 times the number of resolution steps the original solver takes for UNSAT instances). I also suspect NeuRes takes much longer time than NeuroSAT in the proof generation task.  Overall, I don't see the left half of Table 3 adds value to the paper.
>
> Now if we move on to the second half of Table 3 (**Predicted %**). While it is true that NeuroSAT learns to predict satisfiability. This is not its ultimate goal. Therefore, I'm not sure whether it should be used as a strong baseline for the pure task of satisfiabiltiy prediction. To show that NeuRes performs better at satisfiability prediction task than previous method, one should probably consider comparing against methods dedicated for satisfiability prediction (e.g., [1]), on a wider/harder set of benchmarks.
>
> Finally, if we judge the effectiveness of NeuRes from the perspective of a resolution heuristics (which I think we should), I am concerned that the experimental result is not convincing enough. First, the paper only showed resolution step reduction on one simple benchmark set (SR-40); Second, it's not surprising that a more expensive resolution heuristic can reduce the proof length. The key question is whether the increase in time spent on making a heuristic choice is out-weighted by the reduction in the proof length. The paper does not provide a good answer to this question. To me, this paper is very similar in spirit to NeuroCore [2], which tries to reduce the overhead with by only calling the NN heuristic periodically. Perhaps something similar could be explored.
>
> Overall, I find the paper could have done a better job positioning itself in the broader literature of ML + SAT-solving and perform experimental evaluation that more directly validates the effectiveness of the proposed method.
>
> [1] https://ojs.aaai.org/index.php/AAAI/article/view/5733
>
> [2] https://arxiv.org/abs/1903.04671
>
> **Follow up question**: what is the total training time of NeuRes and NeuroSAT, respectively?

---

> > ### Author Response · Authors · 2024-08-13
> >
> > Thank you for your detailed reply, which we address in the following:
> >
> > > The paper seems to position itself as an improved solution over NeuroSAT, while the two methods are incomparable.
> >
> > NeuRes and NeuroSAT do not differ fundamentally in their research direction. While NeuroSAT explores the efficacy of learning a SAT solver by learning satisfiability prediction, NeuRes explores this problem by learning certificate/proof generation. Both approaches are capable of SAT assignment decoding while only NeuRes can prove unsatisfiability.
> >
> > Naturally, they are still comparable on their shared functionalities. In line with our 2nd contribution (lines 66-67), the comparison with NeuroSAT sheds light on the following:
> >
> > 1. Can certificate-based training improve the learnt representations over label-based training? We answer this question positively through our comparison of prediction accuracy of NeuroSAT.
> > 2. Is certificate-based training more data-efficient than label-based training in terms of the number of training samples needed? We answer this question positively through the fact NeuRes was trained on two orders of magnitude fewer samples than NeuroSAT.
> >
> > That is, **{sample+certificate}** contains a much richer learning signal than **{sample+label}**.
> >
> > > Any resolution heuristic can generate a correct proof 100% of the time.
> >
> > The resolution calculus is complete through exhaustive application. We mention this in the introduction (lines 33-36, and 46-47). However, applying resolution naively is practically infeasible since it can easily result in an exponential blow-up in the size of the formula. Indeed, in our evaluation, not a single proof-by-exhaustion case is counted as a successful sample. This is also reflected in the reported $\text{p-Len}$ (ratio between NeuRes and teacher proof lengths). For example, the bootstrapped Full-Attn model has an average $\text{p-Len}$ of 1.15; hence, the proofs are far from the timeout (= 4).
> >
> > > I also suspect NeuRes takes much longer time than NeuroSAT in the proof generation task.
> >
> > NeuroSAT does not generate UNSAT proofs. If you are referring to SAT assignment decoding, then NeuRes is much more time-efficient as it generates a full assignment via a single MLP pass on literal embeddings while NeuroSAT performs an iterative k-means clustering on them (which is significantly costlier and could take hundreds of iterations to converge).
> >
> > > Overall, I don't see the left half of Table 3 adds value to the paper.
> >
> > The left half of that table shows that NeuRes notably outperforms NeuroSAT at finding SAT assignments.
> >
> > > While it is true that NeuroSAT learns to predict satisfiability \[...\] I'm not sure whether it should be used as a strong baseline for the pure task of satisfiabiltiy prediction.
> >
> > We find this argument quite surprising given the fact that NeuroSAT is trained purely as a SAT predictor. The assignment extraction is presented as a desirable byproduct of that learning objective. We used exactly the same prediction method as NeuroSAT to show that the improvement in prediction accuracy is not owed to any variation in the network architecture and can be mainly attributed to a higher quality/informativeness of representations. The prediction comparison serves as a proxy for representation quality.
> >
> > > It's not surprising that a more expensive resolution heuristic can reduce the proof length.
> >
> > The point of this experiment (Section 6.2) is that proof reductions done by bootstrapped NeuRes were learned without any extra supervision, which is fair evidence that NeuRes learns deeper insights into the problem as opposed to simply mimicking the teacher algorithm.
> >
> > > To me, this paper is very similar in spirit to NeuroCore.
> >
> > The NeuroCore paper is about augmenting a state-of-the-art CDCL-based SAT solver with NeuroSAT as a variable selection heuristic (where the *key question* raised by the reviewer would indeed be essential). This is a fundamentally different objective from our paper, which is about improving learned representations by training on proof certificates. Our better representations (from less data) can positively impact approaches like NeuroCore. As mentioned in our initial response to reviewer R9jd, using our representations in a hybrid solver is interesting future work.
> >
> > > Follow up question: what is the total training time of NeuRes and NeuroSAT, respectively?
> >
> > NeuRes takes roughly six days to fully train from scratch on a single NVIDIA A100 GPU (which could be notably sped up by using multiple GPUs). NeuroSAT does not report their total training time, so we do not have this information.
> >
> >
> > We hope our arguments above have convinced the reviewer that the comparisons in our experimental evaluation are carefully chosen. This does require stepping away from looking at our approach through the lens of a resolution heuristic, and instead also valuing the impact of improved representations and data-efficiency on the overall progress of neural methods in this domain.

---

> ### Comment · Reviewer_MVHU · 2024-08-13
>
> I thank the authors for their patient responses and further clarifications. I agree that the discovery that {sample+certificate} contains a much richer learning signal than {sample+label} is significant, and towards this end, this paper provides sufficient empirical data.
>
> However, my concern with the presentation issue remains. *A significant portion of the experimental evaluation--more specifically, half of Section 7 (Proven %) and all of Section 8--is comparing the successful solving rates of the learning-aided DP algorithm and NeuroSAT.* I explained above why this does not make much sense. Perhaps this is partially why I was distracted from the key message the paper is conveying, which, again, I believe is a valuable one.
>
> I increased my score to 4, as I still strongly believe this paper would benefit from a round of revision to address the presentation issues. However, I would not be opposed to this paper being accepted if the other reviewers believe that it is fine to include those not entirely fair comparisons in the paper.

---

### Decision · Program_Chairs · 2024-09-25

**Decision:**

Accept (spotlight)

**Comment:**

In the propositional satisfiability problem, we are given a formula in conjunctive normal form, and the aim is to either determine its satisfiability by providing a satisfying assignment of all variables, or to determine its unsatisfiability by proving that it logically implies the empty clause. In the context of neural approaches to satisfiability, this study focuses on solving both aspects of the problem by deriving correct assignments for satisfiable instances and correct resolution proofs for unsatisfiable ones. To achieve this, the proposed architecture combines a graph neural network with an attention module to simultaneously generate variable assignments and resolution proofs. The key innovation lies in the generation of resolution proofs: at each step, two clauses are selected from attention queries, the resolvant is added to the pool, and the clause embeddings are updated. The paper introduces several attention mechanisms of increasing efficiency. Comparative experiments with NeuroSAT on both satisfiable and unsatisfiable random instances with up to 40 variables are conducted to validate the framework.

Overall, reviewers agree that a key benefit of this learning approach is providing certificates for satisfiable/unsatisfiable instances instead of just predicting a Boolean value. The NeuRes architecture is well-explained, particularly in presenting the different attention mechanisms used for generating resolution proofs. The idea of coupling learning processes for generating assignments and resolution proofs is considered insightful, as these processes can work together to derive a certificate. Additionally, the fact that resolution proofs generated by NeuRes are often shorter than the teacher's proof can be helpful for interpretability.

However, the main weakness of this study may lie in the choice of the competitor in experiments. Some reviewers observed that comparing NeuRes with NeuroSAT is not entirely fair because the latter is unable to prove that an instance is unsatisfiable. A well-argued discussion about the choice of NeuroSAT and the analysis of results would clearly improve the quality of this study. Furthermore, reviewers had questions about ablation studies, comparative model sizes and runtimes, and the gap between NeuRes and highly engineered SAT solvers. Incorporating your detailed responses to these relevant questions in the paper (or appendix) might also improve its quality.